# Clinical Correlates of Osmophobia in Primary Headaches: An Observational Study in Child Cohorts

**DOI:** 10.3390/jcm12082939

**Published:** 2023-04-18

**Authors:** Vittorio Sciruicchio, Daniela D’Agnano, Livio Clemente, Alessandra Rutigliano, Anna Laporta, Marina de Tommaso

**Affiliations:** 1Children Epilepsy and EEG Center, San Paolo Hospital, 70132 Bari, Italy; 2Neurophysiopathology Unit, DiBrain Department, Bari Aldo Moro University, 70121 Bari, Italy

**Keywords:** osmophobia, primary headaches, migraine, childhood, allodynia, disability

## Abstract

Primary headaches, especially migraines, have a significant impact on physical and mental health, as well as on the scholarly performance and quality of life of children and adolescents. Osmophobia could be a potential diagnostic marker of migraine diagnosis and disability. This multicenter observational cross-sectional study included 645 children, aged 8–15, with a diagnosis of primary headaches. We took into consideration the duration, intensity and frequency of headaches, pericranial tenderness, allodynia and osmophobia. In a subgroup of migraine children, we evaluated the migraine-related disability, Psychiatric Self-Administration Scales for Youths and Adolescents, and the Child Version of the Pain Catastrophizing Scale. Osmophobia was found to be present in 28.8% of individuals with primary headaches, with children suffering from migraines having the highest prevalence (35%). Migraine patients with osmophobia also showed a more severe clinical picture, with enhanced disability, anxiety, depression, pain catastrophizing, and allodynia symptoms (F Roy square 10.47 *p* < 0.001). The presence of osmophobia could help in identifying a clinical migraine phenotype coherent with an abnormal bio-behavioral allostatic model that is worthy of prospective observations and careful therapeutic management.

## 1. Introduction

Headaches are a recurrent painful disorder that is widespread in the general population, including children and adolescents. The prevalence of migraine-related headaches in children is thought to be approximately 7–10% (5% in prepuberal children, with percentages increasing throughout adolescence), although diagnostic criteria that are useful for adult headaches may be elusive in many juvenile cases [1,2]. In fact, the clinical characteristics of migraine-related headaches in children can be significantly different from those in adulthood, and features of headaches and the risk of chronic evolution could be misdiagnosed in young patients due to the incompleteness of reported symptoms.

Recently, studies have focused on less frequent symptoms, in particular osmophobia. This symptom is an intolerance to odors and an unpleasant perception that occurs during a headache.

Among the additional anamnestic information, osmophobia constitutes useful information for achieving a correct diagnosis of a migraine [3,4]. In fact, as with primary headaches that occur in adulthood, osmophobia shows a high degree of sensitivity in the diagnosis of migraines in the pediatric population [3,4]. At the developmental age, it is rarely represented in secondary headaches; however, different from adults, it can be present in subgroups of tension-type patients, with a possible evolution toward migraine diagnosis [5,6,7].

Moreover, comparing the prevalence of osmophobia with the symptoms of photo- and phonophobia, we can see that the former is less prevalent. This suggests that only in cases of very intense and painful attacks may the olfactory system be activated [8,9].

Allodynia is a sign of central sensitization, present in adults as well as children with migraines [10], and could be associated with a major risk of chronic evolution and a poor quality of life [11].

In a recent observational study on the prevalence, clinical features and response to preventive treatment in a cohort of primary headache patients with osmophobia selected from a tertiary headache center, we found that this symptom was associated with more severe headache intensity and allodynia [12].

In this observational cross-sectional study, we aimed to evaluate the following:(1)The prevalence of osmophobia in the different forms of primary headaches among a population of children from two headache centers from January 2017 to January 2021.(2)The features of children suffering from headaches and reporting osmophobia in terms of headache frequency, intensity and association with allodynia.(3)The association between osmophobia and migraine-related disability, anxiety, depression, and pain catastrophizing in a subset of children suffering from migraines.

## 2. Materials and Methods

This was an observational cross-sectional study conducted in a global population of 857 children who attended, for the first time, one of two headache centers located in Bari (Puglia, Italy): the headache center of the Neurophysiopathology Unit of Policlinico General Hospital and the headache center of San Paolo Hospital.

In the first center, we took into consideration patients who had attended from January 2017 to January 2021, and in the second center, those from January 2020 to January 2021.

Clinical data were uploaded into a common electronic database.

### 2.1. Inclusion Criteria

The diagnostic criteria were taken from the International Headache Society (HIS) diagnosis guidelines [13]. We included patients with primary headaches aged 8–15. In cases of probable primary headache diagnosis, we only included patients with a confirmed diagnosis at follow-up. We excluded patients with secondary headaches, an intellectual disability or other neuropsychiatric disorders, as well as patients with general medical diseases. We considered primary headache diagnosis groups with at least 30 cases included. In cases with two diagnoses of primary headache (migraine plus tension-type headache or migraine with aura plus migraine without aura), we considered the headache that was more frequent in the last 3 months. No patient was under preventive treatment at the time of first attendance.

### 2.2. Variables

Demographic data and headache characteristics included: inheritance, age of onset, quality and location of headache, frequency of headache episodes in the last 3 months, intensity of headache on a numerical rating scale from 1 to 10 (NRS), headache duration, possible aura symptoms (detailed as visual or sensory symptoms), effect of physical activity, discomfort to scalp and/or neck touching as a symptom of allodynia, and associated symptoms of nausea, vomiting, phono- and photophobia, osmophobia, lacrimation, and red eye. For osmophobia, we asked all patients and/or parents if children had discomfort from perfumes or different fragrances and/or food odors during acute headaches.

We measured the pericranial tenderness score (TTS) using the scale validated by Langermak and Olesen [14] and employed in childhood headaches [10,15].

For the first and second aims, we took into consideration: the duration of headaches, NRS, and frequency of headaches, for which we divided patients into five subgroups (a, less than 1 day with headaches in the last 3 months; b, 1–4 days with headaches; c, 5–10; d, 11–15; e, more than 15 days with headaches), the presence of aura symptoms, the presence of allodynia resulting from the subjective impression of discomfort to the scalp and/or neck touching, and the presence of osmophobia.

For the third aim, apart from the above-reported variables, we evaluated the Pediatric Migraine Disability Score (PedMIDAS) [16,17] together with the Psychiatric Self-Administration Scales for Youths and Adolescents [18] and the Child Version of the Pain Catastrophizing Scale [19] in a subgroup of migraine patients. In these patients, a quantitative analysis of allodynia symptoms was performed. We proposed the same questionnaires employed for adults [20,21], consisting of the symptom checklist reported by Lipton et al. [22]. For allodynia severity, the average number of allodynia symptoms across different migraine attacks was considered [11].

### 2.3. Statistical Analysis

Considering that the literature reports a frequency of osmophobia of around 30% in a total population of about 1000 cases [6,12], we estimated a sample size of 649 patients for α = 0.01 (C.I. 99%) and 516 for α = 0.05 (C.I. 95%).

For the first and second aims, we used the chi-squared test to evaluate the prevalence of osmophobia in the different forms of primary headaches, the distribution of sex, the familial history, the frequency groups and the presence of allodynia among osmophobic and non-osmophobic children. After the estimation of data linearity by means of a Kolmogorov–Smirnov test, the MANOVA analysis served to evaluate how variables, such as headache duration, pericranial tenderness and headache intensity, were different considering the presence of osmophobia and a primary headache diagnosis.

For the third aim, the MANOVA analysis served to evaluate if the PedMIDAS, pain catastrophizing, anxiety and depression scores were different between osmophobic and non-osmophobic migraine children.

## 3. Results

### 3.1. Frequency of Osmophobia and Demographic Data

In Figure 1, we report the flow chart with the patient inclusion criteria. We found osmophobia in 186 of the 645 patients (28.8%). Patients with episodic tension-type headaches presented with a lower prevalence of osmophobia (chi-squared 23.05; DF 5; *p* < 0.001) (Table 1). Female patients dominated within the osmophobia group (59.3% of females in the non-osmophobia group, 69.9% in the osmophobic group; chi-squared 6.37; *p* = 0.012). Age was similar between the two groups (11.51 ± 2.65 not osmophobia; 11.80 ± 2.60 osmophobia; F 1.18; *p* = 0.27). The inheritance factor for headaches was equally distributed between osmophobic and non-osmophobic children (87.7% in osmophobic children, 84.3% in non-osmophobic children; chi-squared 1.47; DF 1; n.s).

### 3.2. Characteristic of Headache Patients with Osmophobia

In terms of frequency, we did not find a different distribution of headache frequency within the groups between patients with or without osmophobia (chi-squared 3.58; DF 4; *p* = 0.46). The MANOVA analysis, including headache features, such as pericranial tenderness, maximal headache intensity evaluated from 0 to 10, headache duration in years, osmophobia and headache diagnosis as factors, showed a difference between osmophobic and non-osmophobic children (F Roy squared 3.34; *p* = 0.019) and among different headache diagnoses (F = 6.76; *p* < 0.001). The interaction between osmophobia and headache diagnosis was not significant (F = 2.16; *p* = 0.091). In detail, headache duration was significantly longer in children reporting acute osmophobia (F = 9.55; *p* = 0.002). Considering headache diagnosis, the intensity of headache was reported as significantly lower in children suffering from episodic tension-type headaches, compared to all migraine groups (F 6.55; *p* < 0.001; Bonferroni: CTE vs. MO, MA, and CM *p* < 0.01) (Table 2). Most of the patients with osmophobia presented with acute symptoms of allodynia (chi-squared 55.12; DF 1; *p* < 0.001) (Figure 2). All five CTH patients with osmophobia had symptoms of allodynia.

### 3.3. Anxiety, Depression, Pain Catastrophizing and Severity of Allodynia in Migraine Patients with Osmophobia

In a subgroup of 183 migraine children (27 MWA, 16 CM and 140 MO), we found that patients with osmophobia (80 subjects) presented with more severe disabilities, as measured with the Pediatric Migraine Disability Assessment (PedMIDAS) (Figure 3), anxiety (Figure 4) and depression (Figure 5), as measured with the Psychiatric Self-Administration Scales for Children and Adolescents (SAFA), pain catastrophizing (Figure 6) and the number of allodynia symptoms (Figure 7) (F Roy square 10.47; *p* < 0.001). The migraine diagnosis accounted for a different expression of symptoms (F 4.25; *p* < 0.001). In particular, the PedMIDAS increased in CM patients compared to the other groups (F 7.93; *p* < 0.01; Bonferroni CM vs. MO and MA, *p* < 0.05) (Figure 3). The interaction between the presence of osmophobia and migraine diagnosis was also significant (F 4.10; *p* = 0.002), as CM and MWA patients with osmophobia had more severe allodynia (*p* = 0.002). (Figure 7).

## 4. Discussion

In this observational study, we confirmed that osmophobia was present among children with primary headaches, prevailing in migraine groups.

It was associated with a longer headache duration, higher headache intensity and the presence of acute allodynia but was equally represented among the different frequency groups.

Migraine children with osmophobia generally had a more severe disease, expressed by higher disability, anxiety, depression, pain catastrophizing scores and the number of allodynia symptoms. In the following paragraphs, we will discuss our aims.

### 4.1. Prevalence of Osmophobia in Primary Headaches

We found that osmophobia was represented in 28.8% of the primary headache population, with predominance in migraine children (around 35% of the total 522 migraine population). A study of 96 primary headaches found a prevalence of 20% [9]; in another study, it was 18.5% among 275 primary headache patients and 25% in migraine children [23]. A multicenter study in a large cohort of juvenile headache cases (1020 cases) indicated a global prevalence of osmophobia in about 28.8% of patients, with about 35% in migraine cases [6], with similar results confirmed in a smaller case series [24]. In 300 patients with primary headaches, Bernardo et al. suggested that osmophobia had 95.8% specificity for migraine diagnosis in children [4], an important element in favor of the prediction of migraine evolution in patients with a diagnosis of probable tension-type headaches [7]. In these studies, osmophobia was investigated in detail using structured interviews and questionnaires. The present study, showing a similar prevalence of osmophobia among a large headache cohort, could confirm that the simple question we included in our clinical record concerning the generic intolerance to perfumes or different fragrances and/or food odors could be sufficiently reliable to detect this symptom in routine clinical practice. Thus far, osmophobia has been found to be present in a consistent portion of juvenile migraine patients and in a subset of chronic tension-type children, as well as in adult primary headaches [12]. In adult primary headaches, we found a higher proportion of osmophobic patients (37.9%), compared to the results in children, including chronic tension-type headaches [12]. This is in line with the association we found with headache duration, as osmophobia may be a symptom appearing in the course of the patient’s headache history. In our pediatric cohort, we had a few patients with confirmed tension-type headache diagnoses. However, the present results in the children assessed mirrored what we observed in adults [12], that osmophobic tension-type headache patients were chronic, expressed more pericranial tenderness and had a higher headache intensity. In accordance with adult headaches and a previous study [4], we can confirm that osmophobia could be a marker of more severe migraines as well as tension-type headaches. In fact, it was associated with a longer headache duration, higher headache intensity and more severe pericranial tenderness, a sign of central sensitization. In fact, most of the patients with osmophobia experienced acute allodynia.

### 4.2. Features of Osmophobic Patients in the General Primary Headache Cohort

Analogous with the results in adult primary headache cohorts [12,25,26,27], we confirmed that osmophobic children, including the few with chronic tension-type headaches, had more severe disease and presented with more signs of central sensitization. Allodynia and pericranial tenderness could indicate the presence of the phenomenon of central sensitization in both adult and childhood headaches [10,28]. Intolerance to odors may be a general sign of hypersensitivity to sensory stimuli, which could predispose the patients to hyperactivity of central circuits devoted to pain processing [29].

The presence of this association—osmophobia plus signs of central sensitization—in childhood headaches could indicate that a predisposition to such phenomena exists [30]. Detecting such a predisposition in early age, also employing easy and feasible questionnaires included in routine clinical reports, could provide aid in the prevention of the chronic evolution of headaches.

### 4.3. Association between Osmophobia and Migraine Disability, Anxiety, Depression and Pain Catastrophizing in a Subset of Migraine Children

The analysis of the clinical correlates of osmophobia in a subset of migraine children confirmed that this symptom marks more severe migraines. In these patients, we applied disability questionnaires, anxiety, depression and pain catastrophizing scores, and quantified symptoms of allodynia using ad hoc questionnaires, as in our previous studies [10]. We found similar associations between osmophobia and higher levels of disability, as well as anxiety, depression and allodynia, similar to what has been observed in adult migraines [12]. Anxious and depressed children could be prone to overestimating sensory stimuli, exhibiting an abnormal bio-behavioral allostatic model, including osmophobia as well as pain hypersensitivity [31,32]. The cortical limbic substrate of mood, pain and smell [33,34] could account for the psychopathological and severe headache profile we found in osmophobic patients. Additionally, the tendency to overestimate the severity of pain, detected with pain catastrophizing, characterized patients with osmophobia. In a previous study, we observed that children with both episodic and chronic migraines exhibited higher scores of pain catastrophizing in association with symptoms of central sensitization and a poor quality of life [35]. Analogous with previous results, osmophobia was associated with signs of severe migraines in both episodic and chronic migraine groups. The evolution of episodic migraines with osmophobia could be a matter for prospective studies in view of a possible transition into chronic evolution.

### 4.4. Limitations

In this study, in a large cohort of primary headache patients, we used a routine and not detailed interview for osmophobia; thus, more patients could have been found with the application of specific questionnaires. However, the prevalence of osmophobia we found among primary headache patients, migraines in particular, was similar to that observed in studies employing specific methods of investigation [6], so the simple investigation of the presence of discomfort to common odors could be enough to relieve this important symptom. The number of tension-type headache cases was not balanced with migraine cases, a problem common in the third-level headache centers, where only patients with severe headaches generally attend. Studies in the general population would be useful to understand the prevalence of osmophobia among less severe headaches, in particular episodic tension-type headaches, and its predictive value for future migraine diagnoses or its evolution into chronic forms.

Authors should discuss the results and how they can be interpreted from the perspective of previous studies and of the working hypotheses. The findings and their implications should be discussed in the broadest context possible. Future research directions may also be highlighted.

## 5. Conclusions

The present study confirmed the high prevalence of osmophobia in migraine children and, in particular, outlined its association with headache severity and symptoms of central sensitization [36]. The simple detection of osmophobia in clinical practice could aid in identifying a clinical phenotype coherent with an abnormal bio-behavioral allostatic model that is worthy of prospective observation and careful therapeutic management.

## Figures and Tables

**Figure 1 jcm-12-02939-f001:**
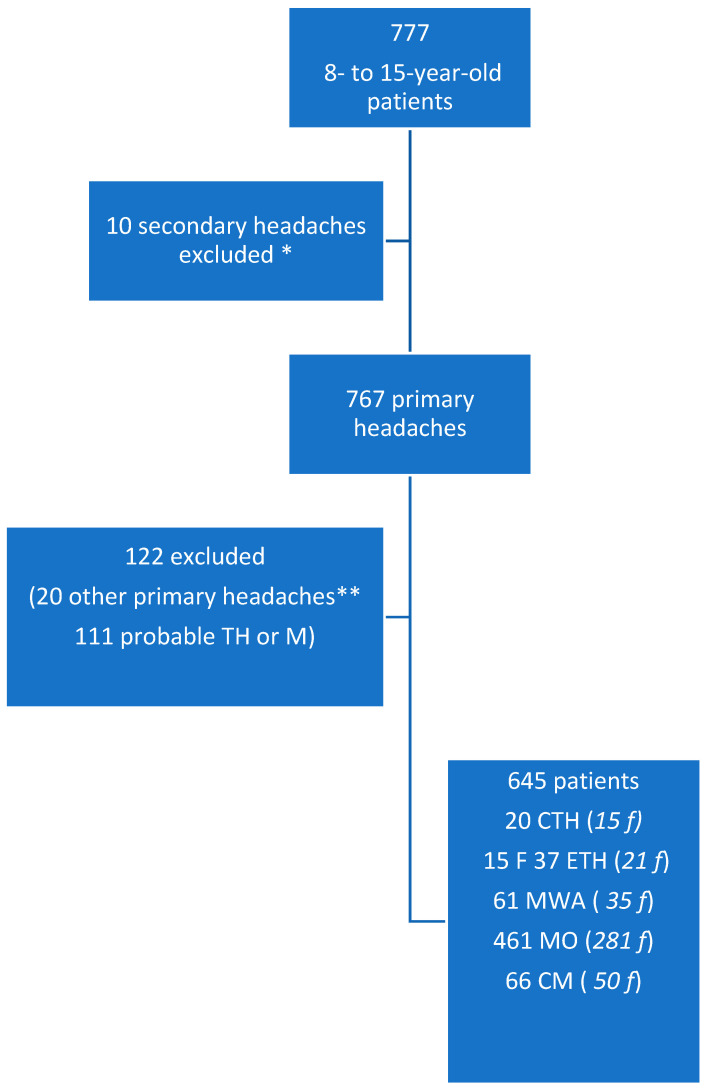
Flow chart depicting patient inclusion criteria. TH: tension-type headache; M: migraine; ETH: episodic tension-type headache; CTH: chronic tension-type headache; MWA: migraine with aura; MO: migraine without aura; CM: chronic migraine; * post-traumatic, cranial hyper- and hypotension, somatoform disease, cranial neuralgia; ** 1 nummular headache, 2 primary exercise headaches, 11 primary stabbing headaches, 2 benign paroxysmal vertigo episodes, 1 abdominal migraine, 2 cyclical vomiting syndrome, 2 new daily persistent headaches.

**Figure 2 jcm-12-02939-f002:**
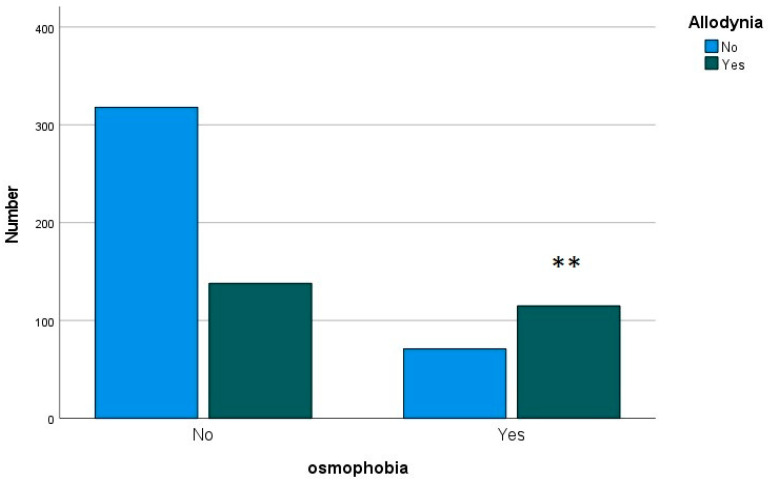
Number of primary headache patients presented with symptoms of acute allodynia in the groups without and with osmophobia. In the latter group, patients with allodynia dominated. ** chi-squared 55.12; *p* < 0.001.

**Figure 3 jcm-12-02939-f003:**
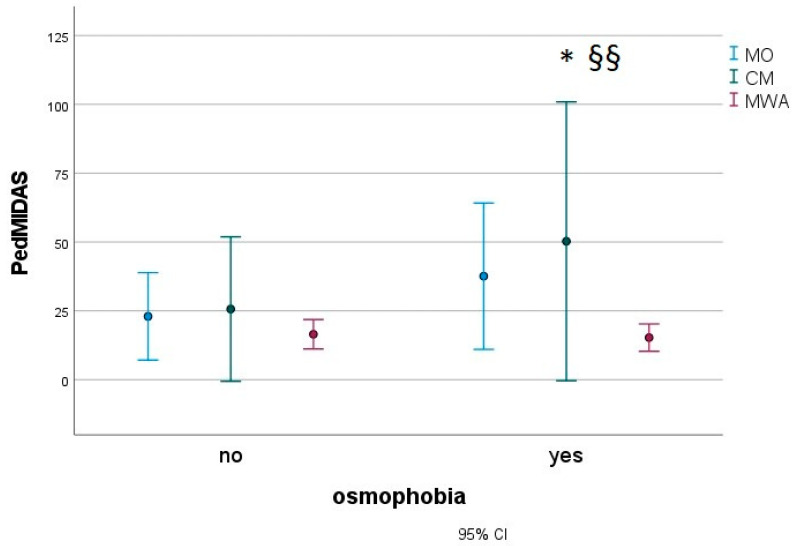
Mean and 95% C.I. of PedMIDAS scores in osmophobic and non-osmophobic migraine children. MWA: migraine with aura; MO: migraine without aura; CM: chronic migraine. Results for osmophobia as a factor * F 4.87, *p* = 0.028; diagnosis as a factor ^§§^ F 7.93, *p* < 0.001; Bonferroni CM vs. MO and MWA, *p* < 0.05).

**Figure 4 jcm-12-02939-f004:**
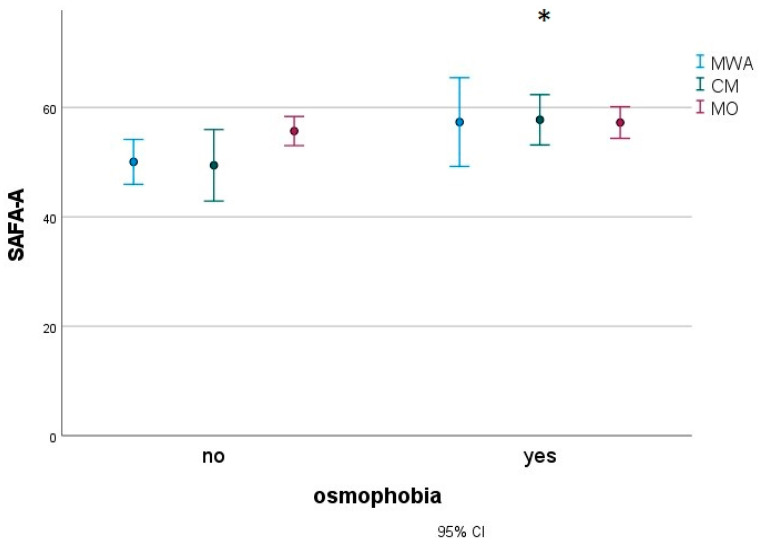
Mean and 95% C.I. of SAFA-Anxiety scores in osmophobic and non-osmophobic migraine children. MWA: migraine with aura; MO: migraine without aura; CM: chronic migraine. Results for osmophobia as a factor * F 4.91, *p* = 0.028.

**Figure 5 jcm-12-02939-f005:**
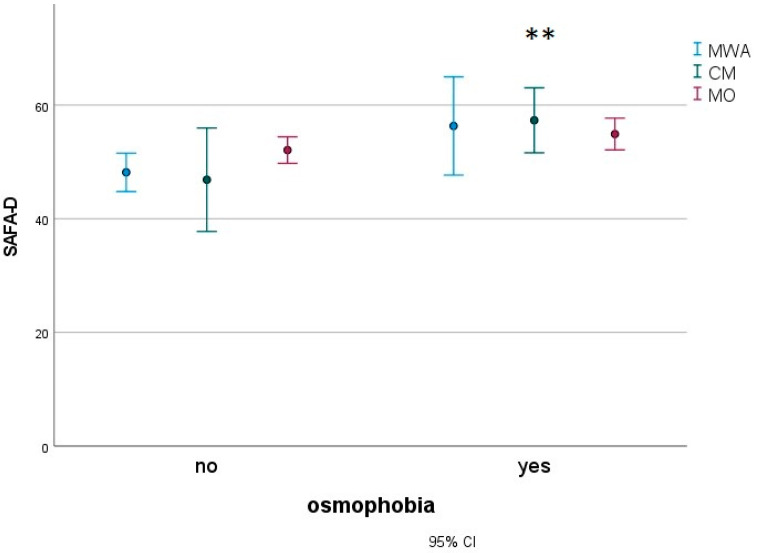
Mean and 95% C.I. of SAFA-Depression scores in osmophobic and non-osmophobic migraine children. MWA: migraine with aura; MO: migraine without aura; CM: chronic migraine. Results for osmophobia as a factor ** F 9.18, *p* = 0.003.

**Figure 6 jcm-12-02939-f006:**
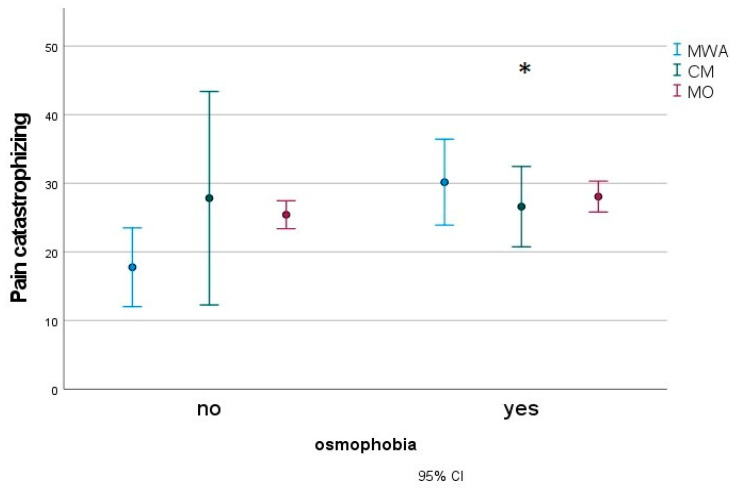
Mean and 95% C.I. of Pain Catastrophizing scores in osmophobic and non-osmophobic migraine children. MWA: migraine with aura; MO: migraine without aura; CM: chronic migraine. Results for osmophobia as a factor * F 4.34, *p* = 0.038.

**Figure 7 jcm-12-02939-f007:**
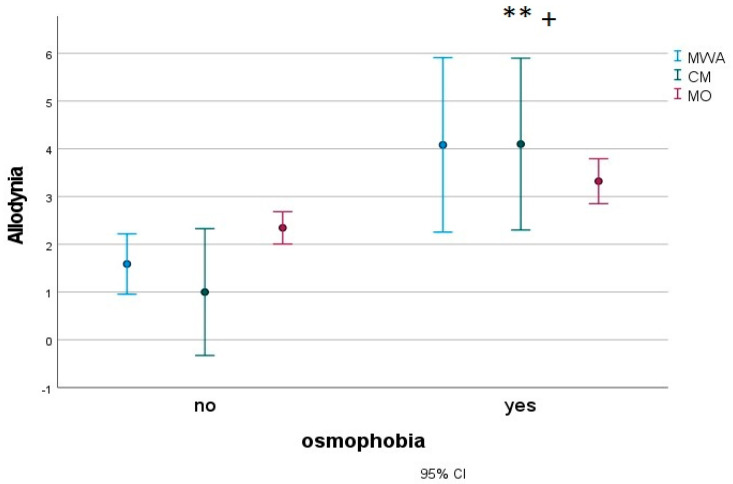
Mean and 95% C.I. of Allodynia scores in osmophobic and non-osmophobic migraine children. MWA: migraine with aura; MO: migraine without aura; CM: chronic migraine. Results for osmophobia as a factor ** F 0.87, *p* = 0.028; ^+^ osmophobia and diagnosis F 6.2, *p* = 0.002.

**Table 1 jcm-12-02939-t001:** Prevalence of osmophobia in the cohort of primary headaches. ETH: episodic tension-type headache; CTH: chronic tension-type headache; MWA: migraine with aura; MO: migraine without aura; CM: chronic migraine. Patients with ETH presented with a lower prevalence of osmophobia (chi-squared *p* < 0.001).

		CTH		ETH		MWA		CM		MO		Total	
		N°	%	N°	%	N°	%	N°	%	N°	%		
Osmophobia	No	15	75	36	97.30	37	60.70	42	63.60	325	70.50	459	71.20
	Yes	5	25	1	2.70	24	39.30	24	36.40	136	29.50	186	28.80

**Table 2 jcm-12-02939-t002:** Headache features in osmophobic and non-osmophobic headache children. In the ETH group, only one child reported osmophobia, so data were omitted. TTS: total tenderness score; NRS: numerical rating scale 0–10 for maximal headache intensity; ETH: episodic tension-type headache; CTH: chronic tension-type headache; MWA: migraine with aura; MO: migraine without aura; CM: chronic migraine. MANOVA for osmophobia F 3.34, *p* 0.019, for headache diagnosis F 6.76, *p* < 0.001; for interaction F 2.16, *p* 0.091. In detail, for osmophobia: headache duration ** F 9.55, *p* 0.002; for headache diagnosis: NRS F 6.55, *p* < 0.001; Bonferroni: CTE vs. MO, MA and CM, *p* < 0.01.

	TTS					NRS					Duration **				
	CTH	ETH	MWA	CM	MO	CTH	ETH	MWA	CM	MO	CTH	ETH	MWA	CM	MO
Osm no	5.00 (1.02)	2.37 (0.73)	1.65 (0.97)	3.50 (0.74)	2.70 (0.29)	6.17 (0.74)	3.91 (0.53)	6.95 (0.70)	6.88 (054)	6.83 (0.21)	2.22 (0.53)	2.17 (0.45)	2.50 (0.6)	3.29 (0.46)	3.08 (0.18)
Osm yes	4.00 (4.34)	.	4.08 (1.25)	4.83 (1.25)	2.67 (0.53)	9.00 (3.12)	.	8.17 (0.9)	5.17 (0.9)	7.15 (0.38)	9.00 (2.67)	.	2.92 (0.77)	4.42 (0.77)	4.26 (0.33)

## Data Availability

Anonymized data are available on request from the corresponding author.

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
