# Peer review of "Clinical Correlates of Osmophobia in Primary Headaches: An Observational Study in Child Cohorts"

_jcm, 2023, doi:10.3390/jcm12082939_

Round 1
Reviewer 1 Report
Dear Authors,
I have read with interest the manuscript and I send you my comments:
1) Statistical analysis: please add the power calculation
2) Figure and tables are very hard to understand please modify these
3) Results: its seems an epidemiological study with few data, please add clinical data for each patient. In clinical practice it is important to define the characteristics of patients with specific symptoms and if or not the patients used a pharmacological treatment. It is not possible to evaluate a differentiation for age , sex and familiarity in the development of headache symptoms
Reviewer 2 Report
About: Clinical correlates of osmophobia in primary headaches: an observational study in children cohorts.
The authors presented interesting data, but I have major concerns on some aspects, namely:
1. Extensive editing of English language and style is required.
2. Sample size was not calculated. Sample size should be calculated based on previous studies about the theme.
3. By the way, more extensive review of recent literature review is required. There are at least two recent studies that should be considered by the authors:
Premonitory and Accompanying Symptoms in Childhood Migraine. Sampaio Rocha-Filho PA, Gherpelli JLD.Curr Pain Headache Rep. 2022 Feb;26(2):151-163. doi: 10.1007/s11916-022-01015-z.
Osmophobia and Odor-Triggered Headaches in Children and Adolescents: Prevalence, Associated Factors, and Importance in the Diagnosis of Migraine. Albanês Oliveira Bernardo A, Lys Medeiros F, Sampaio Rocha-Filho PA.Headache. 2020 May;60(5):954-966. doi: 10.1111/head.13806.
4. In introduction there is the following sentence: Recently studies focused on some less frequent symptoms, especially osmophobia. Which studies?
5. More details about the statistical analysis are needed: significance leve (p), distribution of quantitative data, reason for the selected tests.
6. Figure 1 and Tables 1 and 2 must be reformulated due to inadequate presentation. Other figures and tables should bring some statistical information (teste employed; significance level).